

# Discrimination of Euro 5 gasoline vs. Diesel light-duty engine primary and secondary particle emissions using multivariate statistical analysis of high-resolution mass spectrometry (HRMS) fingerprint

Camille Noblet[1,2], Francois Lestremau[1,3], Adrien Dermigny[1], Nicolas Karoski[1], Claudine Chatellier[1],
Jérôme Beaumont[1], Yao Liu[4], Boris Vansevenant[4], Jean-Luc Besombes[2], and Alexandre Albinet[1]

[1]Institut National de l'Environnement industriel et des RISques (Ineris), 60550 Verneuil en Halatte, France
[2]Université Savoie Mont-Blanc, EDYTEM, 73000 Chambéry, France
[3]HSM, Univ Montpellier, IMT Mines Ales, CNRS, IRD, Ales, France[4]Université Gustave Eiffel, Univ Lyon, AME-EASE, F-69675 Lyon, France

*Correspondence to*: Alexandre Albinet (alexandre.albinet@ineris.fr) and Francois Lestremau (francois.lestremau@mines-ales.fr)

**Abstract.** Emissions from gasoline and Diesel vehicles are predominant anthropogenic sources in ambient air, and their accurate source apportionment is a major concern for air quality policymakers aiming to implement effective strategies to reduce air pollution. Recent studies indicate that particulate matter (PM) emissions from modern cars equipped with the latest
after-treatment technologies are mainly related to secondary organic aerosol (SOA) production, particularly in the case of gasoline vehicles. However, distinguishing in ambient air between emissions from gasoline and Diesel vehicles remains challenging and is rarely achieved. This study aimed to evaluate the potential of non–targeted screening (NTS) analyses for determining specific organic molecular markers of primary organic aerosols (POA) and SOA from gasoline and Diesel vehicles, which could enhance PM source apportionment efforts. Experiments were conducted using a chassis dynamometer
with Euro 5 gasoline and Diesel vehicles under three different driving cycles. Exhaust emissions were diluted before being introduced into a potential aerosol mass oxidation flow reactor (PAM-OFR) to simulate atmospheric aging and SOA formation. Samples were collected both upstream and downstream of the PAM-OFR and analysed using NTS approaches with liquid- and gas-chromatography coupled to quadrupole time-of-flight mass spectrometry (LC- and GC-QToF-MS). The chemical fingerprints obtained were compared using multivariate statistical analyses, including principal component analysis (PCA),
hierarchical clustering analysis (HCA), and partial least square discriminant analysis (PLS-DA). Results revealed specific fingerprints of POA and SOA for each type of vehicle tested and about 10 markers unique to each fraction of Diesel and gasoline vehicles. This study demonstrates the promise of combining high-resolution mass spectrometry based NTS with advanced multivariate statistical analyses to differentiate OA fingerprints and discover specific markers of Diesel and gasoline vehicular sources for further use in PM source apportionment studies.



## 1 Introduction

Vehicular exhaust emissions are a major source of pollutants in ambient air and therefore associated with significant health impacts (Eastwood, 2008; Khomenko et al., 2023; Pant and Harrison, 2013). Significant efforts have been undertaken to reduce polluting emissions, particularly through the implementation of stricter vehicle emission standards, such as the EURO standards. To meet these requirements, several technological advancements have been introduced, including in terms of fuel formulations, combustion processes (e.g., direct injection), and exhaust aftertreatment systems, such as three-way catalytic converters (TWC), selective catalytic reduction (SCR), Diesel oxidation catalysts (DOC), and Diesel particulate filter (DPF). These systems have led to substantial reductions in primary pollutant emissions (Bessagnet et al., 2022; Fiebig et al., 2014; Maricq, 2023). For instance, the introduction of DPFs in EURO 5 Diesel vehicles since 2009 enabled a reduction of more than 90 % in primary particulate emissions (Bergmann et al., 2009; Fiebig et al., 2014; Maricq, 2023). More recently, gasoline vehicles have also been equipped with gasoline particulate filter (GPF) to effectively reduce particulate emissions from gasoline direct injection (GDI) engines (Joshi and Johnson, 2018; Maricq, 2023). However, even modern engines continue to emit significant amounts of volatile and semi-volatile organic compounds (VOCs and SVOCs), which undergo atmospheric (photo)chemical and condensation processes to form secondary organic aerosols (SOA) accounting for a substantial fraction of fine particulate matter (PM) in ambient air.

Several recent smog chamber studies have suggested that the amount of SOA formed in the atmosphere is significantly higher than the primary PM emissions for the most modern vehicles (Chirico et al., 2010; Gentner et al., 2012, 2017; Gordon et al., 2014; Karjalainen et al., 2016; Kuittinen et al., 2021; Robinson et al., 2007; Tkacik et al., 2014). This effect is particularly pronounced in gasoline engines compared to modern Diesel vehicles equipped with DPF (Gentner et al., 2012, 2017; Hartikainen et al., 2023; Kostenidou et al., 2024; Platt et al., 2017). Thus, to develop effective strategies for reducing air pollution, it is crucial to distinguish between emissions from gasoline and Diesel vehicles, as well as between primary and secondary particulate emissions, to accurately evaluate their respective contributions to ambient air PM levels.

Source-receptor models are widely used to estimate the contributions of various source types, including vehicular emissions (Hopke et al., 2020). However, only a limited number of studies have attempted to distinguish between emissions from gasoline and Diesel vehicles. Most of these studies rely on the chemical mass balance (CMB) approach, utilizing traditional aerosol chemical speciation such as elemental carbon (EC), organic carbon (OC), major ions, and metals, or incorporating a set of organic species like polycyclic aromatic hydrocarbons (PAHs), hopanes, cholestanes, n-alcanes, etc. (Al-Naiema et al., 2018; Gertler, 2005; Heo et al., 2013; Srimuruganandam and Nagendra, 2012; Wang et al., 2012). Nevertheless, the common used source profiles of gasoline and Diesel vehicular emissions remain highly uncertain and should be regarded as approximations (Lough et al., 2007). Indeed, selecting representative profiles in CMB for modern vehicle emissions can be challenging (Bray et al., 2019; Hopke et al., 2020; Karagulian et al., 2015; Pernigotti et al., 2016; Simon et al., 2010; Viana et al., 2008). While positive matrix factorization (PMF) models mostly gave a global factor for vehicular emissions due to the lack of specific source markers (tracers) and often similar temporal patterns (Dallmann et al., 2014; Wang et al., 2017; Wang and Hopke, 2013;



Watson et al., 1994), some studies used of a combination of chemical speciation with some VOCs (propane, butane, pentane...) to better quantify the contributions of gasoline and Diesel vehicles (Lambe et al., 2009; Thornhill et al., 2010; Wong et al., 2020). However, these approaches remain underutilized and are often challenging to apply in urban environments with multiple contributing sources. The distinction between gasoline and Diesel emissions in source-receptor models is further complicated by the shared emission of many inorganic and organic compounds, and the lack of specific markers unique to either source. Recent studies have suggested 1-nitropyrene as a promising marker for Diesel emissions (Keyte et al., 2016; Schulte et al., 2015; Zielinska et al., 2004) and some authors have attempted to incorporate it in PMF source apportionment (Lanzafame et al., 2021; Srivastava et al., 2018, 2019, 2021). Finally, even though SOA becomes the dominant fraction of organic aerosols emitted by modern vehicles, the identification of specific SOA markers for gasoline or Diesel emissions remains unachieved to date.

The emergence of high-resolution mass spectrometry (HRMS) has revolutionized the study of OA by enabling the acquisition of highly accurate and detailed mass data (Laskin et al., 2018; Nizkorodov et al., 2011; Nozière et al., 2015). The application of non-targeted analysis (NTS) strategies with HRMS has facilitated the detection of hundreds to thousands of unique compounds in ambient air or combustion emissions, driving significant research efforts to highlight novel pollutants without prior knowledge (Avagyan et al., 2016; Manz et al., 2023; Röhler et al., 2020, 2021; Vogel et al., 2019; Xu et al., 2021). Despite this progress, relatively few studies have fully exploited the potential of untargeted HRMS analyses to determine specific markers or chemical signatures for OA source apportionment purposes. By combining HRMS data with multivariate statistical techniques such as principal component analysis (PCA) or partial least squares–discriminant analysis (PLS-DA), NTS strategies allow the comparison of chemical fingerprints obtained from different experimental conditions and potentially discover key substances of interest. Few examples are described in the literature demonstrating the utility of this approach for pinpointing markers, or chemical fingerprints, specific of primary or secondary OA sources (Huo et al., 2021; Mu et al., 2019; Noblet et al., 2024; Thoma et al., 2022; Weggler et al., 2016).

The objective of this work was to assess the potential of NTS analyses to reveal discriminating chemical features (markers) of primary organic aerosols (POA) and SOA from gasoline and Diesel vehicles. Experiments have been conducted using a DPF-equipped Diesel vehicle and a GDI vehicle. Both vehicles have been operated over ambient- and hot-start driving cycles to simulate typical urban driving conditions. The diluted emissions have been photo-oxidized in a potential aerosol mass - oxidation flow reactor (PAM-OFR) to investigate the formation of secondary PM. Collected samples were analysed using both liquid- and gas-chromatography high-resolution mass spectrometry (LC-HRMS and GC-HRMS), enabling comprehensive profiling of polar and nonpolar organic compounds (Haglund et al., 2024; Huo et al., 2021; Moschet et al., 2017, 2018; Rostkowski et al., 2019). The resulting data were processed using computational tools and multivariate statistical analyses to compare the chemical signatures and reveal potential chemical features (markers) for differentiating between primary/secondary emissions from the two vehicular sources.



## 2 Material and methods

### 2.1 Vehicle tested, driving cycles and fuels

A EURO 5 GDI vehicle and a EURO 5 Diesel vehicle were utilized in this study. The gasoline vehicle was equipped with a TWC (but without GPF), while the Diesel vehicle featured a DOC and a DPF as part of its exhaust after-treatment systems. Additional vehicle details are provided in Table S1 in the Supplementary Material (SM). Both vehicles were tested on a chassis dynamometer under three driving cycles: the worldwide harmonized light vehicles test cycle (WLTC) (Tutuianu et al., 2015) and the common Artemis driving cycles (CADC) for urban and motorway (MW) driving conditions (Andre, 2004; Andre et al., 2006). Speed-time profiles for these cycles are shown in Fig. S11–S13.

To replicate real-world driving conditions and account for variability in emissions, different engine starting conditions were used. WLTC cycles were conducted under ambient-start temperature conditions (first cycles of the day), whereas CADC cycles (urban and motorway) were performed with warmed-up engines. Each vehicle completed the ambient-start WLTC cycle twice, while both hot-start CADC cycles were repeated four times. Since vehicle emissions are highly influenced by driving conditions, particularly engine start-up temperatures (Chirico et al., 2010; Karjalainen et al., 2016; Kuittinen et al., 2021; Yusuf and Inambao, 2019), two consecutive CADC cycles with warmed-up engines were conducted to collect sufficient material for chemical analysis. All tests were conducted using commercial gasoline and Diesel that adhered to summertime fuel standards. Additional information on fuel composition is provided in Table S2.

### 2.2 Sampling setup

Figure 1 illustrates the experimental setup used in this study. Each vehicle was operated on a chassis dynamometer according to the various tested driving cycles. Exhaust emissions were collected directly from the tailpipe using a thermally insulated and externally heated exhaust transfer line maintained at 120 °C. The exhaust was then diluted using a Dekati Fine Particle Sampler 4000 (FPS) system, with the primary dilution temperature set to 120 °C. The total dilution ratios were calculated from $CO_2$ measurements both at the emission source and after dilution. For the gasoline vehicle, dilution ratios ranged from 18 to 43, while for the Diesel vehicle, they ranged from 16 to 30.

A Potential Aerosol Mass-Oxidation Flow Reactor (PAM-OFR) was installed downstream of the dilution system to evaluate secondary aerosol formation by exposing emissions to hydroxyl radicals (OH) throughout the driving cycles (Kang et al., 2007; Lambe et al., 2011). The PAM-OFR was operated at a flow rate of approximately 9.0–9.5 L min$^{-1}$, corresponding to an average residence time of 84–88 s. The temperature inside the PAM-OFR ranged from 25 to 35 °C for the Diesel vehicle and 26 to 30°C for the gasoline vehicle. Vehicular emissions were humidified using a Nafion membrane humidifier (Permapure LLC, FC100-80 Series, 15 cm, stainless steel), achieving relative humidity levels inside the PAM-OFR of 37-52 % for the Diesel vehicle and 48–50 % for the gasoline vehicle. OH radicals were generated using four UV lamps ($\lambda$ = 185 and 254 nm). The PAM-OFR operated in OFR185 mode (Li et al., 2015; Peng and Jimenez, 2020), where $O_2$ is photolyzed at 185 nm to produce $O_3$, which is then photolyzed at 254 nm to generate $O(^1D)$ that reacted with $H_2O$ to produce OH radicals. During all



experiments, the lamp voltage was set to 2.5 V, and the irradiation, measured via a photodiode inside the PAM-OFR, remained constant. OH exposure was determined during spare experiments by continuously measuring the decay of $SO_2$ (200 ppb) using an $SO_2$ analyser (AF 21 M, Environnement S.A.), following the method outlined by Lambe et al. (2011). For the gasoline

vehicle, average OH exposures ranged from $7.9 \times 10^{11}$ to $1.6 \times 10^{12}$ molecules cm$^{-3}$ s, depending on the driving cycle (Figure 2). For the Diesel vehicle, OH exposures were lower, ranging from $1.4 \times 10^{11}$ to $1.1 \times 10^{12}$ molecules cm$^{-3}$ s, with the lowest values observed during the CADC MW cycle due to the presence of high $NO_x$ concentrations in the exhaust emissions (Figure 2). These OH exposure levels align with values reported by Cao et al. (2020) under similar flow rate conditions within the PAM-OFR.  Equivalent photochemical aging time was estimated assuming an average atmospheric OH concentration of $1 \times$

$10^6$ molecules cm$^{-3}$ (global 24-h average; Finlayson-Pitts and Pitts, 2000). For the gasoline vehicle, the equivalent photochemical age ranged from 6 to 10 days, while for the Diesel vehicle, it ranged from 0.9 to 7 days. To minimize background levels, the PAM-OFR was systematically flushed overnight with treated ambient air (total filter + activated charcoal scrubber) and all lamps at 10 V. Additionally, the exhaust sampling line was replaced between the tests of the two vehicles to avoid cross-contamination.

For each vehicle, fresh and photochemically aged PM emissions were collected on quartz filters (Pall Flex, Tissuquartz, Ø = 47 mm, pre-combusted at 500 °C for 12 hours to remove organic contaminants) over the entire driving cycles. Background filter samples (field blanks) were obtained during separate dynamic blank experiments (performed without any vehicle under the same conditions as the driving cycles. In total, 32 PM samples were collected including 12 samples for each vehicle and eight field blanks. After collection, PM samples were individually stored in 47 mm crystal polystyrene Petri dishes, sealed in

polyethylene bags, and kept at −18°C until analysis.

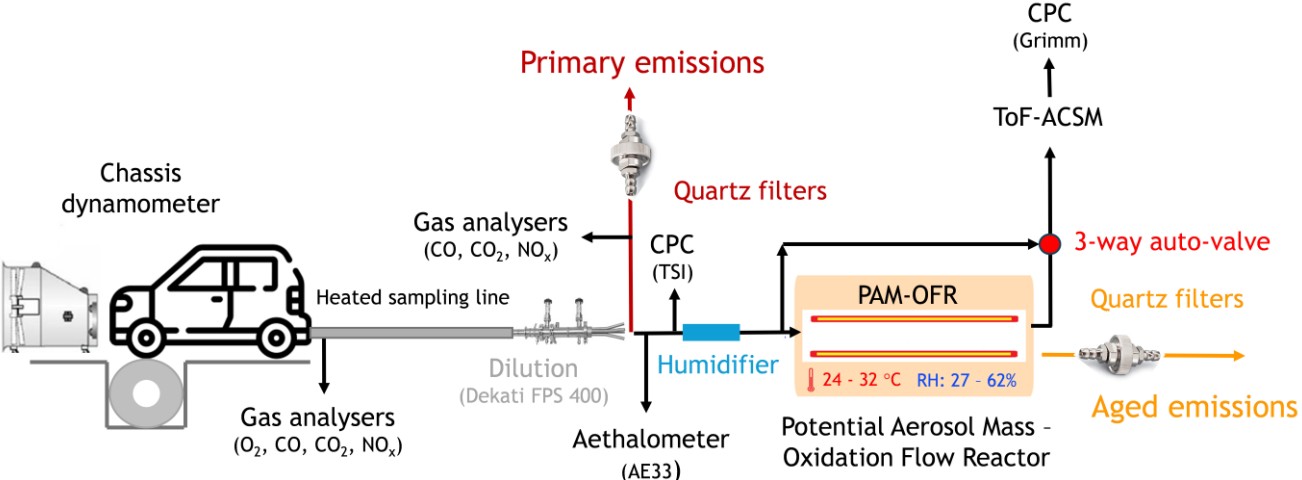

**Figure 1: Schematic of the experimental setup used for the EURO 5 gasoline and Diesel vehicle emissions and aging tests.**





## 2.3 Instrumentation and sampling

Table S3 provides a summary of the instruments used to monitor the primary and aged emissions during the experiments. Continuous monitoring (at one minute resolution) was conducted for gaseous emissions such as oxygen ($O_2$), carbon monoxide (CO), carbon dioxide ($CO_2$), and nitrogen oxides ($NO_x$) both, at the emission source and after dilution. An ozone monitor (Model 202, 2B Technologies) was used to measure ozone concentrations inside the PAM-OFR, which ranged from 4.3 to 5.5 ppm for the gasoline vehicle and from 0.2 to 3.4 ppm for the Diesel vehicle. Particle instrumentation was positioned

downstream of the dilution system. Black carbon (BC) mass concentrations of primary particles were continuously monitored using a multi-wavelength aethalometer (AE33, Magee Scientific) with a 1-minute time resolution. Primary and secondary particle number concentrations were measured using Condensation Particle Counters (CPC, Grimm Serie 5.400 for primary emissions, and TSI 3775 for aged emissions) at a one second time resolution. Aerosol chemical composition was analysed using a Time-of-Flight Aerosol Chemical Speciation Monitor (ToF-ACSM, Aerodyne) equipped with a $PM_1$ aerodynamic lens

and operated at a 40 second time resolution. Detector response factor calibration was performed with ammonium nitrate and sulphate solutions (Crenn et al., 2015; Freney et al., 2019; Ng et al., 2011). A relative ionization efficiency (RIE) of 1.4 was applied for organic matter (OM), and a collection efficiency (CE) of 0.5 was used for the entire ACSM dataset. No corrections were applied for lens transmission. To sample primary or aged emissions, an electrically actuated three-way valve was employed. Background OM and inorganic species mass concentrations were subtracted from ToF-ACSM measurements using

field dynamic blanks (experiments in the same conditions but with the vehicle engine off). Emission factors (EF) for both primary and secondary OM were calculated using Eq. (1):

$$EF_{OM} = \frac{[OM] \times 1000 \times DF \times EFR \times TD}{D \times FC} \tag{1}$$

with EFOM: OM emissions factor in mg kg-1 of fuel burnt,

[OM]: organics measured by the ToF-ACSM in µg m$^{-3}$,

DF: dilution factor determined with $CO_2$ measurements,

EFR: exhaust flow rate in m$^3$ min$^{-1}$,

TD: test duration in min,

D: driven distance in km,

FC: fuel consumption in kg km$^{-1}$.

## 2.4 Chemical analyses

### 2.4.1 Sample extraction

Details of the solvents and chemicals used for the extraction procedures, including their purity and suppliers, are described in Supplementary Material (Table S4). Prior to extraction, samples were spiked with labelled internal standards, referred to as extraction internal standards (EIS), to monitor the performance of the entire sample analysis process. These labelled





compounds were selected to be distributed across the chromatogram of analysis. The solution contained five EIS for LC-QToF analysis (for both positive and negative ionization modes) and eight EIS for GC-QToF analysis (Table S5). A QuEChERS-like (Quick, Easy, Cheap, Effective, Rugged, and Safe) procedure was developed and optimized for specific extractions for LC- and GC-HRMS analyses (Albinet et al., 2013, 2014, 2019). In brief, 47 mm diameter quartz filter samples, including matrix blanks (calcined filters only), were extracted with 6 mL of acetonitrile via agitation for 5 minutes using a multi-position

vortex (1700 rpm) (Multi-tube Vortex, DVX-2500, VWR). Prior to extraction a volume of 30 μL of each EIS solution was spiked onto each filter. After extraction, the samples were centrifuged for 7 minutes at 4500 rpm (Sigma 3–16 PK centrifuge). The supernatant (approximately 4 mL) was then collected and filtered (Uptidisc PTFE 13 mm, 0.2 μm).

Before solvent evaporation, the supernatants from the same driving cycle tests and aging state (primary and aged emissions) were pooled to ensure sufficient material for chemical analysis. The pooled sample extracts were evaporated under a nitrogen

flow to a final volume of 200 μL, with the tubes immersed in a thermostated bath set at 45 °C. This process yielded 12 final sample extracts of 200 μL each, comprising six samples per vehicle, half representing primary emissions and the other half aged emissions, with one sample per driving cycle. Additionally, one quality control (QC) sample was prepared by pooling 30 μL from each sample extract under investigation. Final extracts were stored at −20 °C until analysis.

### 2.4.2 NTS chemical analyses

Analytical methods were optimized to maximize chromatographic separation and minimize interferences during detection. Prior to analysis, internal injection standards (IIS) were added to the extracts to assess the performance of the instrumental analysis (Table S6). The LC-QToF analyses were performed using a UHPLC system (1290 Infinity, Agilent) with a C18 column (Acquity HSST3 C18, 2.1 mm × 100 mm, 1.8 μm, Waters) equipped with a guard column (Acquity UPLC HSS T3 VanGuard, 2.1 × 5 mm, 1.8 μm, Waters). The assembly was maintained in an oven at a thermostatically controlled temperature

of 40 °C. Elution was conducted using methanol (mobile phase B) and water containing 1 mM acetic acid and 1 mM ammonium acetate (mobile phase A) at a flow rate of 0.4 mL min$^{-1}$. The injection volume was 5 μL, and the gradient conditions are detailed in Table S7. Following chromatographic separation, detection was carried out using a quadrupole-time of flight mass spectrometer (QToF IFunnel 6550, Agilent) equipped with an electrospray ionization (ESI) source operating in both positive (ESI (+)) and negative (ESI (−)) modes. Instrumental parameters for the QToF are provided in Table S8. The GC-

QToF analyses were conducted using an Agilent 7890 gas chromatograph equipped with a DB-5MS column (30 m × 250 μm, 0.25 μm, Agilent J&W) and a guard column (VF-5ms, 10 m × 0.25 mm, 0.25 μm, Agilent J&W). A 1 μL volume of each extract was injected in pulsed splitless mode using an ultra-inert low-fritted liner (870 μL, 4 mm, Agilent Technologies). Detection was performed with an Agilent 7250 accurate-mass quadrupole QToFMS instrument featuring an electron ionization source, operating at high energy (70 eV, HEI) and low-energy (10 eV, LEI). Operating conditions for the GC-ToF analysis are

detailed in Table S9. To ensure mass accuracy, both instruments were calibrated before each analytical sequence using manufacturer-provided solutions. For GC-QToF, calibrations were also performed every two injections throughout the

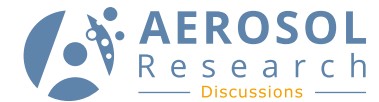

sequence to minimize deviations, achieving precision in measured mass (< 2 ppm). For LC-QToF, a reference solution was continuously injected during the run to monitor instrument drift and ensure analytical reliability.

## 2.5 Quality control

To ensure the reliability of the non-targeted analyses and maintain data quality, several QA/QC measures were implemented using QC pools, extraction internal standards (EIS), and injection internal standards (IIS). Following the guidelines recommended by (Broadhurst et al., 2018) for non-target screening analyses, samples (including lab and field blanks) were analyzed in a randomized sequence. This sequence included injecting ten QC pooled samples at the start to "condition" the analytical system, followed by regular injections of QC pools every five samples throughout the sequence to evaluate and

monitor potential instrument drift. After the analyses, the retention times (RT) and masses of all internal standards (EIS and IIS) were specifically monitored in the QC pooled samples to assess RT stability and QToF-MS mass accuracy. Minor RT drifts for EIS and IIS were observed but remained within acceptable tolerances: ± 0.2 minutes for LC-QToF data and ± 0.1 minutes for GC-QToF data (Figures S2–S7). Additionally, acceptable deviations in the m/z ratio (< 10 ppm) were observed for EIS and IIS in both GC- and LC-QToF data. Peak shapes and areas of the EIS in the QC pools were also monitored.

Reproducibility and instrumental drift in the analytical responses were evaluated using control charts for three selected EIS. The logarithms of the peak areas for these EIS were plotted against the injection order (Figures S8–S10). For each EIS in the QC pools and vehicular samples, the mean area ($\overline{x}$) and standard deviation ($\sigma$) were calculated. Most values fell within the range $\overline{x} \pm 2\sigma$. Any values outside this range were specifically examined. However, no trends or systematic issues were identified. Consequently, no data were excluded.

## 230   2.6 Data validation, data treatment and multivariate statistical analyses

The Recursive feature extraction module in the Profinder software (Agilent v. B.10.00) was used to process three-dimensional LC raw data (retention time, m/z ratio, and abundance) and two-dimensional GC raw data (retention time and abundance) into aligned chromatographic peaks with corresponding peak abundances. Detailed parameter settings, along with the number of detected and selected features in the final dataset, are presented in Table S10. Features with more than 30 % missing values in

QC samples were excluded. In total, 1088, 1833, and 1779 features from GC-QToF, LC-ESI(+)-QToF, and LC-ESI(−)-QToF data, respectively, were retained for subsequent multivariate statistical analyses.

Before statistical analysis, missing values, caused by undetected features or signals below the threshold limit, were replaced with a non-significant value (half of the minimum area found in the entire dataset). Feature abundances were then normalized using pooled QC samples (Di Guida et al., 2016), log-transformed, and scaled by autoscaling. Multivariate statistical analyses

were conducted using MetaboAnalyst software (Chong et al., 2019). Non-supervised methods, such as PCA and agglomerative hierarchical clustering analysis (HCA), were performed to explore associations among the OA samples from both vehicles. For HCA, classification was based on the Pearson correlation coefficient and the average linkage method.





PLS-DA was used to identify specific markers or chemical signatures associated with Diesel and gasoline vehicles for both primary and secondary emissions. Model quality was assessed using leave-one-out cross-validation (LOOCV) with two parameters: $R^2(Y)$ (goodness-of-fit) and $Q^2(Y)$ (goodness-of-prediction). The $R^2X$ and $Q^2(Y)$ values are summarized in Table S11. Variable importance in projection (VIP) scores were used to select compounds with the highest discrimination potential. To ensure the reliability of highlighted markers, each chemical entity was systematically verified in raw data for presence/absence across groups, abundance levels, and chromatographic responses (e.g., resolution and peak shape).

### 2.7 Tentative identification

Marker identification was not the primary aim of this work; however, preliminary attempts to identify the highlighted markers were carried out and further explored in a dedicated study. Molecular formulas of the selected markers were proposed based on LC-QToF data using MassHunter software. These formulas were estimated from the $[M+H^+]$ or $[M-H^-]$ ions, considering isotope patterns (abundance and spacing) and low mass error limits ($\Delta m/z \leq 10$ ppm). Based on available literature about elements present in fuels and lubricants, formula assignments were performed under the following constraints: $C \leq 50$, $H \leq 100$, $N \leq 10$, $O \leq 20$, $S \leq 5$, $Al \leq 5$, $Zn \leq 5$, $Cd \leq 5$, $Ce \leq 5$, $Co \leq 5$, $Ca \leq 5$, $Pd \leq 5$, $Si \leq 5$, $Mn \leq 5$, $Pb \leq 5$, and $Ni \leq 5$ (Allen et al., 2001; Harrison et al., 2003; Hu et al., 2009; Huang et al., 1994; Maricq et al., 2002; R'Mili et al., 2018; Rogge et al., 1993). In cases of multiple assignments, only molecular formulas with a score higher than 90 (empirical score based on exact mass, isotope spacing, and abundance) were considered.

For GC-QToF data, features were extracted using the Agilent Unknowns Analysis program (version B.09.00) with peak detection and grouping performed via the SureMass deconvolution algorithm. The parameters used were an RT window size factor of 80 and an extraction window $\Delta m/z$ delta of $\pm 0.3$ AMU (left) and $\pm 0.7$ AMU (right). For each deconvoluted spectrum, a forward search (pure weight factor = 0.7) was conducted using the NIST17 mass spectra library. A match factor was assigned based on spectral similarities and retention index agreement. Retention indices were calculated using a solution of C8 to C40 n-alkanes. The candidates obtained from NIST17 were manually reviewed.

## 3 Results and discussion

### 3.1 Overview of the particulate and gaseous emissions of EURO 5 Diesel and gasoline vehicles

Time series data for the WLTC and CADC (urban and motorway) driving cycles (Figures S10–S12) indicated that gaseous emissions were strongly influenced by driving conditions, aligning with previous findings in the literature (Alves et al., 2015; Zervas and Bikas, 2008). $NO_x$ concentrations were particularly sensitive to high-speed conditions and to the frequency and intensity of acceleration and deceleration phases. These variations are likely driven by the increased combustion temperatures and elevated air–fuel ratios associated with such operating conditions. Ambient start conditions had a notable impact on CO emissions, primarily due to suboptimal combustion during cold starts. At low engine and catalyst temperatures, incomplete fuel oxidation occurs because the combustion process is less efficient, and the catalytic converter has not yet reached its light-



off temperature. Exhaust particle number concentrations were largely higher for the GDI engine, without GPF, compared to
the DPF-equipped Diesel vehicle. This was consistent with previous observations for similar vehicle types (Kostenidou et al.,
2021; Louis et al., 2017; Maricq, 2023; Mathis et al., 2005), since GPF started being implemented on GDI vehicles for the
Euro 6 norm. Like gaseous emissions, primary particle emissions were closely correlated with driving conditions. For warm
cycles (CADC urban and MW), the highest total primary particle number concentrations were observed during acceleration
phases and at high speeds. Ambient start driving conditions (WLTC) resulted in four times higher total primary particle number
concentrations, especially for a short period after starting (Drozd et al., 2016; Kostenidou et al., 2021). The temporal variations
of the main chemical species of the primary particles followed the same trends, as shown for the WLTC driving cycle (Figures
S13 and S14), where OM accounted for the main fraction of the exhaust particles Figure 2 compares to the average OM
emission factors (EFs) for both primary emissions and aged emissions from EURO 5 vehicles across different driving cycles.
Primary OM EFs were approximately two to three times higher for the EURO 5 gasoline vehicle than for the EURO 5 DPF-
equipped Diesel vehicle (Figure 2). Similarly, primary BC EFs were 10 to 20 times higher for the gasoline vehicle than for the
Diesel vehicle (Figure S16). These findings suggest that while DPF-equipped Diesel vehicles emit relatively few primary
particles (BC or organics), GDI vehicles without GPF are responsible for substantial BC and organic emissions (Bessagnet et
al., 2022; Gordon et al., 2014; Kostenidou et al., 2021; Maricq, 2023; Maricq et al., 2012). After aging, a substantial formation
of OM (SOA + aged POA) was observed and OM EFs were 8 to 15 times higher for the gasoline vehicle than for Diesel one
(Figure 2, S13 and S14). The influence of ambient start conditions on OM emissions was more pronounced for aged emissions
than for primary emissions, with secondary OM EFs three to six times higher for the WLTC cycle compared to other driving
cycles. Note that the formation of secondary OM was mainly observed during the low-speed phase of the WLTC driving cycle
(Figures S13 and S14). In the case of the Diesel vehicle, elevated $NO_x$ emissions observed during the CADC motorway cycle
resulted in lower average OH exposure, leading to reduced SOA production compared to other driving cycles.





**Figure 2. Diesel and gasoline Euro 5 OM average emissions factors for the primary (POA) and secondary (SOA + aged POA) fractions, average NOx concentrations after dilution exhaust and average OH exposures obtained with the ambient start WLTC, hot-start CADC motorway (MW) and urban driving cycles. The error bars correspond to the standard deviation (± 2 σ) for the number of experiments performed (n = 2 to 4).**

295

300



The results obtained under the tested conditions revealed that secondary organic aerosol (SOA) formation was approximately one to nine times higher for the EURO 5 Diesel vehicle, increasing from 0.2–1.2 to 0.5–6.2 mg kg$^{-1}$ fuel. For the EURO 5 gasoline vehicle, a notable increase in POA emissions was observed, with emission factors rising by a factor of 11 to 22 (from 1.3–3.0 to 13.8–65.9 mg kg$^{-1}$ fuel). These values are consistent with previously reported data. For instance, Karjalainen et al. (2016) found that gasoline SOA emissions after oxidation in a potential aerosol mass oxidation flow reactor (PAM-OFR) were approximately 13 times higher than POA emissions. Although higher SOA emission factors (200 - 400 mg kg$^{-1}$ fuel) were reported by Tkacik et al. (2014) during PAM-OFR experiments in a road tunnel, their SOA-to-POA emission factor ratios were in a similar range (10 to 20 times). Jathar et al. (2017), using PAM-OFR to simulate photochemical aging of Diesel exhaust from vehicles equipped with aftertreatment systems, observed SOA emission factors from 80 to 800 times higher than POA. Smog chamber studies have shown similar trends. For example, Platt et al. (2013) reported SOA emission factors of approximately 350 mg kg$^{-1}$ fuel, about 14 times higher than POA emissions (24.5 mg kg$^{-1}$ fuel) for a EURO 5 gasoline vehicle. Overall, these studies consistently indicate that SOA contributions from modern vehicular emissions far exceed POA levels (Chirico et al., 2010; Gentner et al., 2017; Gordon et al., 2014; Hartikainen et al., 2023; Jathar et al., 2017; Karjalainen et al., 2016; Kostenidou et al., 2024; Kuittinen et al., 2021; Platt et al., 2017; Tkacik et al., 2014). These findings underscore the importance of further characterizing the secondary fraction of vehicular emissions.

### 3.2 Non–target chemical characterization of vehicular OA

### 3.2.1 Vehicular OA chemical signature comparison

To investigate potential correlations in the chemical signatures of vehicular OA samples, PCA score plots were generated based on GC-QToF (Figure 3) and LC-QToF (Figure S17) datasets acquired for both primary and secondary emissions across the tested driving cycles (WLTC ambient start, and motorway and urban CADC with warmed-up engines). For each analytical technique (GC-QToF, LC-ESI(+)-QToF, and LC-ESI(−)-QToF), the corresponding PCA revealed tightly clustered pooled QC samples, indicating high data reliability and reproducibility. The score plots demonstrated a clear separation between POA and SOA samples along the first principal component (PC1). Specifically, POA samples from both Diesel and gasoline vehicles contributed negatively to PC1, while SOA samples contributed positively. Although less pronounced, a partial differentiation between Diesel and gasoline vehicles was also observed for both POA and SOA samples. PC1 appeared to primarily capture the (trans-)formation state of the OA, distinguishing between primary and secondary emissions, while also reflecting, to a lesser extent, the type of vehicle (fuel type) as a source. The second principal component (PC2) captured greater intra-group variability, particularly among the Diesel-derived samples. This variation may be linked to the lower and more inconsistent emissions observed from the DPF-equipped Diesel vehicle, which result in lower particulate mass and, consequently, more variability in chemical signatures of the collected samples.



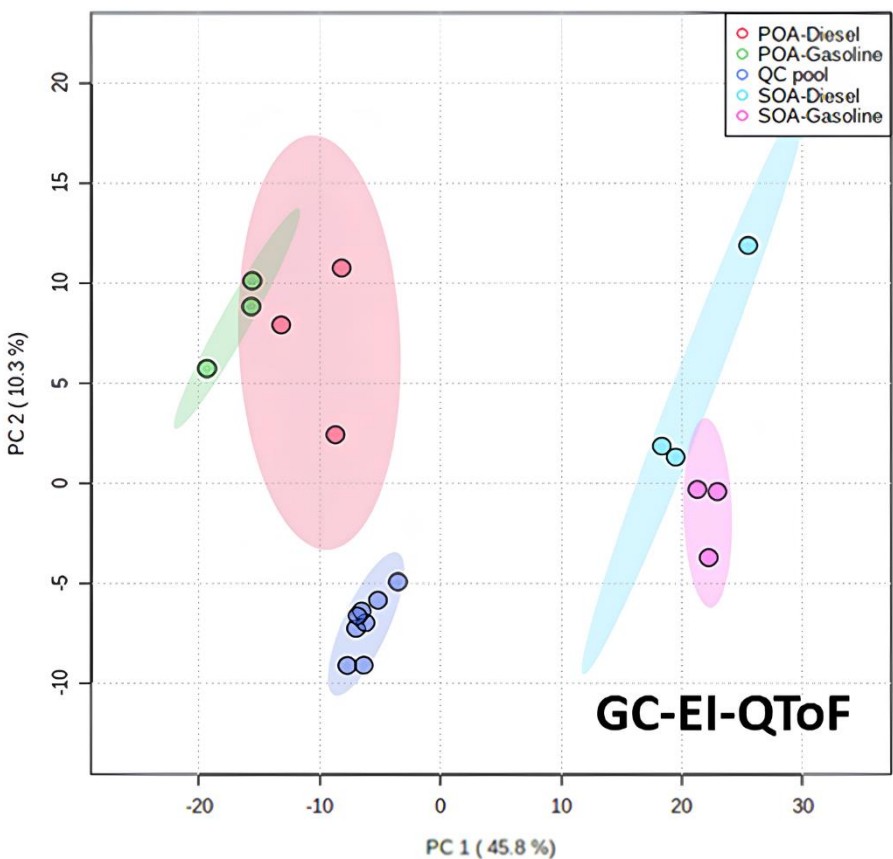

**Figure 3. Principal component analysis (PCA) score plots of samples from primary and secondary vehicular emissions (POA Diesel: red, POA gasoline: green, SOA Diesel: light blue, SOA gasoline: pink) and pooled QC samples (dark blue). One pooled sample per driving cycle tested (ambient start WLTC, hot-start CADC motorway and urban cycles) driving cycles is included. The result is obtained from the NTS analysis performed by GC-QToF. The data have been normalized by PQN (pooled QC samples), log-transformed and auto-scaled. The ellipses represent the 95 % confidence zones.**

For each analytical method employed HCA corroborated the PCA findings, revealing three distinct clusters corresponding to pooled QC samples, POA, and SOA vehicular emissions (Figures 4, S18 and S19). Across all three techniques, HCA clearly distinguished between POA and SOA clusters, highlighting marked differences in feature patterns and further confirming the significant chemical differentiation between primary and aged emissions. These observations are consistent with prior studies (Gentner et al., 2017; Presto et al., 2014; Suarez-Bertoa et al., 2015) which reported a predominance of reduced hydrocarbon species ($C_xH_y$) in primary emissions and a greater abundance of oxygenated compounds in secondary emissions. Beyond the POA-SOA separation, additional sub-clustering of samples was observed based on fuel type. Specifically, GC-QToF and LC-QToF (negative ionization mode) data showed a partial but evident differentiation between gasoline- and Diesel-derived emissions within both POA and SOA clusters. In contrast, LC-QToF data acquired in positive ionization mode did not yield a clear classification between vehicle types, suggesting a less pronounced chemical divergence in the ionizable compounds detected under those conditions.



**Figure 4. Two-way hierarchical clustering and heat map of the different vehicular exhaust samples of normalized GC-QToF data.**
**This classification was performed based on the Pearson's correlation coefficient using the average linkage method. The colour-scale**
**on the right represents the feature relative abundance in each sample relatively compared to the others.**

### 3.2.2 Diesel and gasoline POA chemical signature comparison

PCA were conducted specifically on POA samples from gasoline and Diesel vehicles, using GC-QToF and LC-QToF datasets
(Figure S19). The PCA score plots revealed clustering patterns based on both driving conditions and fuel type, covering



ambient-start WLTC, hot-start CADC motorway and urban cycles. For all three PCA analyses, the first two principal components explained over 60 % of the total variance. PC1 primarily separated Diesel from gasoline POA samples, suggesting a strong influence of fuel type, while PC2 appeared more related to driving cycle conditions. For both vehicle types, PC2 consistently differentiated ambient-start WLTC and hot-start CADC motorway (MW) samples from hot-start CADC urban samples, except in the case of Diesel samples analysed by GC and LC in negative mode, where PC2 generally separated the

hot-start CADC motorway (MW) samples from the others. Notably, LC-positive mode data for the gasoline vehicle indicated that hot-start CADC motorway and ambient-start WLTC samples exhibited more similar chemical signatures compared to hot-start CADC urban samples. These results suggest that chemical signatures of POA are more sensitive to driving dynamics, such as speed and acceleration/deceleration patterns, than to start-up conditions (ambient vs. hot).

    To improve sample classification, PLS-DA models were built, showing robust fit and predictive power (Table S11). The top

30 discriminating features (defined by median m/z–retention time for LC-QToF and retention time for GC-QToF) had VIP scores > 2 for GC data and > 3 for LC data (Figures 5 and S21). These high-VIP entities were predominantly associated with gasoline samples for GC and LC-positive mode data, while features from Diesel samples were more prominent in LC-negative mode. A stringent filtering process was applied to identify reliable chemical markers. A feature was retained only if it was exclusively present in one vehicle type and detected consistently across all three replicates. Table S12 summarizes the selected

discriminating features. For GC data, one marker was identified for each vehicle type. For LC-positive mode, 11 markers were retained for gasoline and one for Diesel. In LC-negative mode, eight markers were retained for Diesel and one for gasoline. Chromatographic profiles and mass spectra (GC data) of selected potential markers are shown in Fig. S23, S24, S27, and S28.






**Figure 5. Partial least square–discriminant analysis (PLS-DA) of POA (top) and SOA (bottom) samples from gasoline and Diesel vehicles. The results are obtained from the GC-QToF data and were normalized by pooled QC samples, log-transformed and auto-scaled. The ellipses represent the 95 % confidence zones. Classification of chemical entities (left scale: retention time) characteristic of each vehicular source according to the VIP score are displayed on the right. The colour scale indicates the variation in abundance of the chemical entity (100 % = red, 0 % = blue) in all samples of both vehicles. Only the first 30 chemical entities with the highest VIP scores are shown on the graph.**




### 3.2.3 Diesel and gasoline SOA chemical signature comparison

For all three PCA analyses corresponding to the different analytical techniques, the first two principal components accounted for more than 60 % of the total variance (Figure 20). In the GC-QToF and LC-QToF negative mode data, PC1 predominantly

separated Diesel and gasoline SOA samples, indicating a distinction in their chemical signatures. This separation was less pronounced in the LC-QToF positive mode data, suggesting that the differences between Diesel and gasoline SOA compositions may be less marked than those observed for POA samples. PC2 consistently separated the Diesel hot-start CADC motorway driving cycle samples from the others across all three analytical methods. This observation aligns with the elevated $NO_x$ concentrations measured during this driving cycle, which likely resulted in emissions with a lower degree of oxidation.

The high $NO_x$ levels may have also triggered distinct atmospheric reactions, leading to the formation of nitrogen-containing compounds. In contrast, no clear trend emerged in the PCA clustering of gasoline SOA samples with respect to driving cycle or start-up conditions (ambient vs. hot).

The 30 top-ranking discriminant features from the PLS-DA models, defined by median molecular mass /retention time for LC-QToF data and retention time for GC-QToF data, with VIP scores > 2 (GC-QToF) and >3 (LC-QToF), are presented in Fig. 5

and S22. For GC-QToF data, PLS-DA models exhibited both strong goodness-of-fit and predictive performance (Table S11). Following rigorous manual review of the raw data, two features specific to gasoline SOA and one feature for Diesel SOA were confirmed (Table S12). Corresponding chromatographic responses and mass spectra are shown in Fig. S29 to S31. In the case of LC-QToF data, although the SOA models demonstrated good model fit, their predictive performance was lower than for POA, particularly in positive ionization mode (Table S11). Ultimately, ten discriminatory entities were selected, with eight

specifics to Diesel SOA (all in negative mode) and two to gasoline SOA (Table S12). Representative chromatographic responses for selected LC-QToF potential markers are illustrated in Fig. S25 and S26.

### 3.3 Tentative identifications of POA and SOA markers

For the GC-QToF data, only one marker compound specific to Diesel POA emissions could be tentatively identified: 2H-Pyran-2-one (Table S12). This compound has not previously been reported as a potential marker of Diesel exhaust and may

be formed through high-temperature reactions catalysed by palladium complexes or acids (Larock et al., 1999). For the remaining POA and SOA markers, no confident identification could be achieved due to the absence of significant spectral matches in the NIST mass spectral database (Table S12). In contrast, LC-QToF data allowed for tentative elemental formula assignments for several features, suggesting the presence of metals and metalloids such as calcium (Ca), aluminium (Al), cobalt (Co), silicon (Si), sulfur (S), and manganese (Mn). The detection of such elements is consistent with earlier findings

highlighting the substantial contribution (ranging from 20% to 80%) of fuel additives and lubricating oils to POA emissions, especially in Diesel vehicles (Alam et al., 2018; Brandenberger et al., 2005; Carbone et al., 2019; Kleeman et al., 2008; Lyu et al., 2024; Sonntag et al., 2012; Worton et al., 2014). Lubricating oils have also been widely recognized as major sources of semi-volatile organic compounds (SVOCs), which can act as important precursors to SOA formation in vehicular emissions

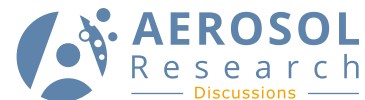

(Ghadimi et al., 2023; Hartikainen et al., 2023; Karjalainen et al., 2019; Zhao et al., 2017). Specifically, Karjalainen et al.

(2019) emphasized the dominant role of lubricating oil in SOA precursor formation from Diesel engines compared to GDI vehicles. Additionally, sulphur-containing organic compounds (e.g., CHOS and CHONS species) are commonly formed during the photochemical aging of gasoline exhaust, as $SO_2$, emitted in trace amounts primarily from the lubricating oil, undergoes atmospheric reactions (Schneider et al., 2024). These observations support the tentative identification of the SOA gasoline marker LC NEG SOA G-1, which likely contains sulphur functionality (Table S12). To further assess the validity of

the proposed molecular formula assignments for the potential markers identified using LC-QToF data, a Van Krevelen diagram was employed (Figure 6). The diagram plots hydrogen-to-carbon (H:C) versus oxygen-to-carbon (O:C) atomic ratios, allowing visualization of the chemical characteristics of the identified compounds. As anticipated, SOA markers exhibited higher O:C ratios compared to POA markers, reflecting the greater degree of oxidation associated with aging processes. However, a few questionable molecular formulas were identified (highlighted in Fig. 6 and listed in Table S12) which fall outside the typical

elemental composition patterns and are likely erroneous. These outliers should therefore be excluded from further interpretation.

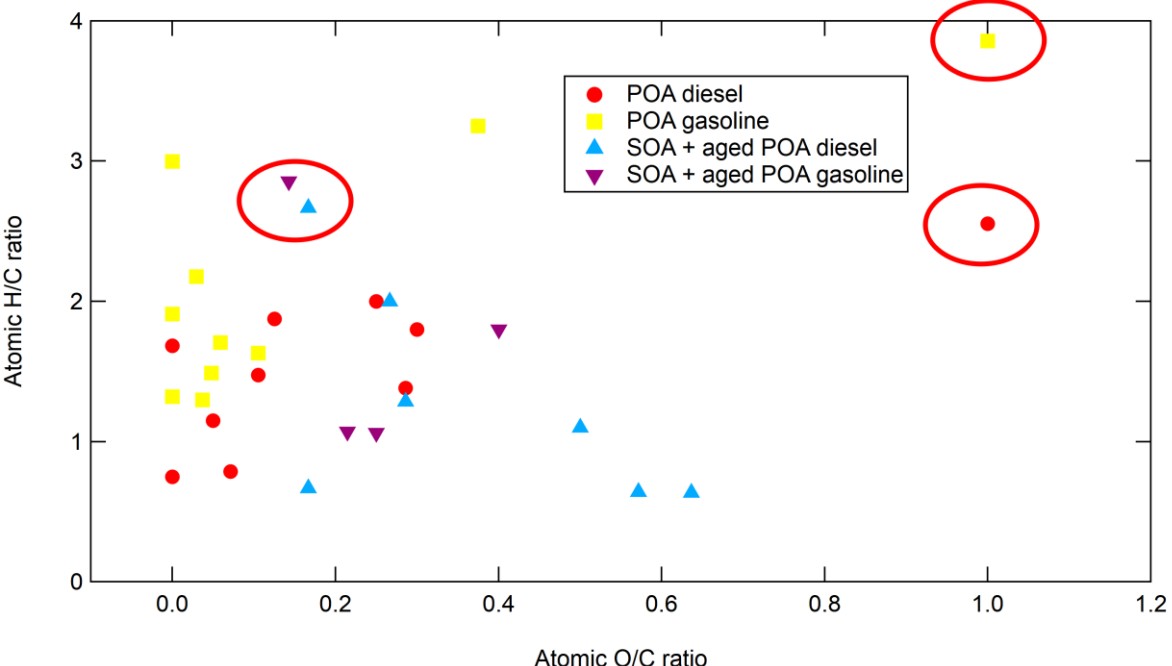

**Figure 6. Van Krevelen diagram representing all potential molecular formulas assigned to the proposed markers obtained from LC-QToF data. Questionable molecular formulas are highlighted with red circles.**

**4 Conclusions**

The potential of NTS strategies to differentiate vehicular OA sources was evaluated using both LC- and GC-HRMS, combined with multivariate statistical analyses, including both unsupervised and supervised approaches. The use of three complementary

analytical techniques enabled a broad chemical characterization of the OA. Although the number of samples analysed was limited, the consistent outcomes observed across all three datasets strengthened the robustness and reliability of the findings.

Unsupervised analyses revealed clear distinctions between POA and SOA emissions from vehicular exhaust, with each fraction exhibiting distinct chemical signatures. Additionally, although to a lesser extent, differences between gasoline and Diesel OA were observed for both POA and SOA. Supervised modelling using PLS-DA enabled the identification of approximately 10 discriminating markers for each vehicle type and emission fraction. While these findings are promising and demonstrate the potential of non-targeted HRMS approaches in source apportionment of vehicular OA, further investigations are necessary to

confirm and generalize the results. Future studies should incorporate a larger number of vehicles including those equipped with the most modern aftertreatment technologies (Diesel with $NO_x$ traps, gasoline GDI with GPF), increased sampling replicates, and a broader range of oxidation conditions to enhance statistical power and environmental relevance.

**Data availability**

The raw data can be provided upon request from the authors.

**Competing interests**

The authors declare that they have no conflict of interest.

**Author contributions**

CN: Data curation, Formal analysis, Investigation, Methodology, Validation, Visualization, Writing – original draft, Writing – review & editing. FL: Conceptualization, Investigation, Methodology, Resources, Supervision, Validation, Writing – review

& editing, Validation. AD: Investigation, Methodology, Writing – review & editing. NK: Investigation, Methodology, Writing – review & editing. CC: Investigation, Methodology, Writing – review & editing, Validation. JB: Investigation, Methodology, Validation, Writing – review & editing. YL: Investigation, Methodology, Resources, Writing – review & editing. BV: Investigation, Methodology, Writing – review & editing. JLB: Conceptualization, Supervision, Writing – review & editing, Validation. AA: Conceptualization, Formal analysis, Funding acquisition, Investigation, Methodology, Project Administration,

Resources, Supervision, Validation, Visualization, Writing – review & editing.

**Acknowledgements**

This work has been supported by the French Ministry of environment as well as by the National reference laboratory for air quality monitoring in France (LCSQA). Collection of the samples has been done taking advantage of the EVORA research project funded by the ADEME agency (French Environment and Energy Management Agency, convention number

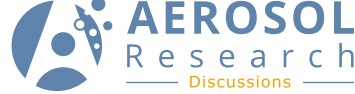

1766C0022). The authors thank Patrick Tassel and Pascal Perret from Université Gustave Eiffel for the vehicle testing experiments on chassis dynamometer. They also thank Agilent for the complimentary loan of the GC-Q-ToF-MS. Finally, authors gratefully acknowledge Bruno Le Bizec, Gaud Dervilly and Yann Guitton for helpful discussions and advice regarding NTS approaches.

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

Contributions to cities' ambient particulate matter (PM): A systematic review of local source contributions at global level,
Atmos. Environ., 120, 475–483, https://doi.org/10.1016/j.atmosenv.2015.08.087, 2015.

Karjalainen, P., Timonen, H., Saukko, E., Kuuluvainen, H., Saarikoski, S., Aakko-Saksa, P., Murtonen, T., Bloss, M., Dal
Maso, M., Simonen, P., Ahlberg, E., Svenningsson, B., Brune, W. H., Hillamo, R., Keskinen, J., and Rönkkö, T.: Time-
resolved characterization of primary particle emissions and secondary particle formation from a modern gasoline passenger
car, Atmospheric Chemistry and Physics, 16, 8559–8570, https://doi.org/10.5194/acp-16-8559-2016, 2016.

Karjalainen, P., Rönkkö, T., Simonen, P., Ntziachristos, L., Juuti, P., Timonen, H., Teinilä, K., Saarikoski, S., Saveljeff, H.,
Lauren, M., Happonen, M., Matilainen, P., Maunula, T., Nuottimäki, J., and Keskinen, J.: Strategies To Diminish the Emissions



of Particles and Secondary Aerosol Formation from Diesel Engines, Environ. Sci. Technol., 53, 10408–10416, https://doi.org/10.1021/acs.est.9b04073, 2019.

Keyte, I. J., Albinet, A., and Harrison, R. M.: On-road traffic emissions of polycyclic aromatic hydrocarbons and their oxy- and nitro-derivative compounds measured in road tunnel environments, Sci. Total Environ., 566, 1131–1142, https://doi.org/10.1016/j.scitotenv.2016.05.152, 2016.

Khomenko, S., Pisoni, E., Thunis, P., Bessagnet, B., Cirach, M., Iungman, T., Barboza, E. P., Khreis, H., Mueller, N., Tonne, C., Hoogh, K. de, Hoek, G., Chowdhury, S., Lelieveld, J., and Nieuwenhuijsen, M.: Spatial and sector-specific contributions of emissions to ambient air pollution and mortality in European cities: a health impact assessment, The Lancet Public Health, 8, e546–e558, https://doi.org/10.1016/S2468-2667(23)00106-8, 2023.

Kleeman, M. J., Riddle, S. G., Robert, M. A., and Jakober, C. A.: Lubricating Oil and Fuel Contributions To Particulate Matter Emissions from Light-Duty Gasoline and Heavy-Duty Diesel Vehicles, Environ. Sci. Technol., 42, 235–242, https://doi.org/10.1021/es071054c, 2008.

Kostenidou, E., Martinez-Valiente, A., R'Mili, B., Marques, B., Temime-Roussel, B., Durand, A., André, M., Liu, Y., Louis, C., Vansevenant, B., Ferry, D., Laffon, C., Parent, P., and D'Anna, B.: Technical note: Emission factors, chemical composition, and morphology of particles emitted from Euro 5 diesel and gasoline light-duty vehicles during transient cycles, Atmospheric Chemistry and Physics, 21, 4779–4796, https://doi.org/10.5194/acp-21-4779-2021, 2021.

Kostenidou, E., Marques, B., Temime-Roussel, B., Liu, Y., Vansevenant, B., Sartelet, K., and D'Anna, B.: Secondary organic aerosol formed by Euro 5 gasoline vehicle emissions: chemical composition and gas-to-particle phase partitioning, Atmospheric Chemistry and Physics, 24, 2705–2729, https://doi.org/10.5194/acp-24-2705-2024, 2024.

Kuittinen, N., McCaffery, C., Peng, W., Zimmerman, S., Roth, P., Simonen, P., Karjalainen, P., Keskinen, J., Cocker, D. R., Durbin, T. D., Rönkkö, T., Bahreini, R., and Karavalakis, G.: Effects of driving conditions on secondary aerosol formation from a GDI vehicle using an oxidation flow reactor, Environmental Pollution, 282, 117069, https://doi.org/10.1016/j.envpol.2021.117069, 2021.

Lambe, A. T., Logue, J. M., Kreisberg, N. M., Hering, S. V., Worton, D. R., Goldstein, A. H., Donahue, N. M., and Robinson, A. L.: Apportioning black carbon to sources using highly time-resolved ambient measurements of organic molecular markers in Pittsburgh, Atmos. Environ., 43, 3941–3950, https://doi.org/10.1016/j.atmosenv.2009.04.057, 2009.

Lambe, A. T., Ahern, A. T., Williams, L. R., Slowik, J. G., Wong, J. P. S., Abbatt, J. P. D., Brune, W. H., Ng, N. L., Wright, J. P., Croasdale, D. R., Worsnop, D. R., Davidovits, P., and Onasch, T. B.: Characterization of aerosol photooxidation flow reactors: heterogeneous oxidation, secondary organic aerosol formation and cloud condensation nuclei activity measurements, Atmospheric Measurement Techniques, 4, 445–461, https://doi.org/10.5194/amt-4-445-2011, 2011.



Lanzafame, G. M., Srivastava, D., Favez, O., Bandowe, B. A. M., Shahpoury, P., Lammel, G., Bonnaire, N., Alleman, L. Y.,
Couvidat, F., Bessagnet, B., and Albinet, A.: One-year measurements of secondary organic aerosol (SOA) markers in the Paris
region (France): Concentrations, gas/particle partitioning and SOA source apportionment, Science of The Total Environment,
757, 143921, https://doi.org/10.1016/j.scitotenv.2020.143921, 2021.

Larock, R. C., Doty, M. J., and Han, X.: Synthesis of Isocoumarins and α-Pyrones via Palladium-Catalyzed Annulation of
Internal Alkynes, J. Org. Chem., 64, 8770–8779, https://doi.org/10.1021/jo9821628, 1999.

Laskin, J., Laskin, A., and Nizkorodov, S. A.: Mass Spectrometry Analysis in Atmospheric Chemistry, Anal. Chem., 90, 166–
189, https://doi.org/10.1021/acs.analchem.7b04249, 2018.

Li, R., Palm, B. B., Ortega, A. M., Hlywiak, J., Hu, W., Peng, Z., Day, D. A., Knote, C., Brune, W. H., de Gouw, J. A., and
Jimenez, J. L.: Modeling the Radical Chemistry in an Oxidation Flow Reactor: Radical Formation and Recycling, Sensitivities,
and the OH Exposure Estimation Equation, J. Phys. Chem. A, 119, 4418–4432, https://doi.org/10.1021/jp509534k, 2015.

Lough, G. C., Christensen, C. G., Schauer, J. J., Tortorelli, J., Mani, E., Lawson, D. R., Clark, N. N., and Gabele, P. A.:
Development of Molecular Marker Source Profiles for Emissions from On-Road Gasoline and Diesel Vehicle Fleets, Journal
of the Air & Waste Management Association, 57, 1190–1199, https://doi.org/10.3155/1047-3289.57.10.1190, 2007.

Louis, C., Liu, Y., Martinet, S., D'Anna, B., Valiente, A. M., Boreave, A., R'Mili, B., Tassel, P., Perret, P., and André, M.:
Dilution effects on ultrafine particle emissions from Euro 5 and Euro 6 diesel and gasoline vehicles, Atmospheric Environment,
169, 80–88, https://doi.org/10.1016/j.atmosenv.2017.09.007, 2017.

Lyu, X., Liang, X., Wang, Y., Wang, Y., Zhao, B., Shu, G., Tian, H., and Wang, K.: Influence of lubricants on particulate
matter emission from internal combustion engines: A review, Fuel, 366, 131317, https://doi.org/10.1016/j.fuel.2024.131317,
2024.

Manz, K. E., Feerick, A., Braun, J. M., Feng, Y.-L., Hall, A., Koelmel, J., Manzano, C., Newton, S. R., Pennell, K. D., Place,
B. J., Godri Pollitt, K. J., Prasse, C., and Young, J. A.: Non-targeted analysis (NTA) and suspect screening analysis (SSA): a
review of examining the chemical exposome, J Expo Sci Environ Epidemiol, 33, 524–536, https://doi.org/10.1038/s41370-
023-00574-6, 2023.

Maricq, M. M.: Engine, aftertreatment, fuel quality and non-tailpipe achievements to lower gasoline vehicle PM emissions:
Literature    review    and    future    prospects,    Science    of    The    Total    Environment,    866,    161225,
https://doi.org/10.1016/j.scitotenv.2022.161225, 2023.

Maricq, M. M., Chase, R. E., Xu, N., and Laing, P. M.: The Effects of the Catalytic Converter and Fuel Sulfur Level on Motor
Vehicle Particulate Matter Emissions: Light Duty Diesel Vehicles, Environ. Sci. Technol., 36, 283–289,
https://doi.org/10.1021/es010962l, 2002.

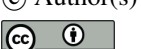


Maricq, M. M., Szente, J. J., and Jahr, K.: The Impact of Ethanol Fuel Blends on PM Emissions from a Light-Duty GDI
Vehicle, Aerosol Science and Technology, 46, 576–583, https://doi.org/10.1080/02786826.2011.648780, 2012.

Mathis, U., Mohr, M., and Forss, A. M.: Comprehensive particle characterization of modern gasoline and diesel passenger cars
at low ambient temperatures, Atmos. Environ., 39, 107–117, https://doi.org/10.1016/j.atmosenv.2004.09.029, 2005.

Moschet, C., Lew, B. M., Hasenbein, S., Anumol, T., and Young, T. M.: LC- and GC-QTOF-MS as Complementary Tools
for a Comprehensive Micropollutant Analysis in Aquatic Systems, Environ. Sci. Technol., 51, 1553–1561,
https://doi.org/10.1021/acs.est.6b05352, 2017.

Moschet, C., Anumol, T., Lew, B. M., Bennett, D. H., and Young, T. M.: Household Dust as a Repository of Chemical
Accumulation: New Insights from a Comprehensive High-Resolution Mass Spectrometric Study, Environ. Sci. Technol., 52,
2878–2887, https://doi.org/10.1021/acs.est.7b05767, 2018.

Mu, M., Li, X., Qiu, Y., and Shi, Y.: Study on a New Gasoline Particulate Filter Structure Based on the Nested Cylinder and
Diversion Channel Plug, Energies, 12, 1–19, 2019.

Ng, N. L., Herndon, S. C., Trimborn, A., Canagaratna, M. R., Croteau, P. L., Onasch, T. B., Sueper, D., Worsnop, D. R.,
Zhang, Q., Sun, Y. L., and Jayne, J. T.: An Aerosol Chemical Speciation Monitor (ACSM) for Routine Monitoring of the
Composition and Mass Concentrations of Ambient Aerosol, Aerosol Science and Technology, 45, 780–794,
https://doi.org/10.1080/02786826.2011.560211, 2011.

Nizkorodov, S. A., Laskin, J., and Laskin, A.: Molecular chemistry of organic aerosols through the application of high
resolution mass spectrometry, Phys. Chem. Chem. Phys., 13, 3612–3629, https://doi.org/10.1039/C0CP02032J, 2011.

Noblet, C., Lestremau, F., Collet, S., Chatellier, C., Beaumont, J., Besombes, J.-L., and Albinet, A.: Aerosolomics based
approach to discover source molecular markers: A case study for discriminating residential wood heating vs garden green
waste burning emission sources, Chemosphere, 352, 141242, https://doi.org/10.1016/j.chemosphere.2024.141242, 2024.

Nozière, B., Kalberer, M., Claeys, C., Allan, J. D., D'Anna, B., Decesari, S., Finessi, E., Glasius, M., Grgić, I., Hamilton, J.
F., Hoffmann, T., Iinuma, Y., Jaoui, M., Kahnt, A., Kampf, C. J., Kourtchev, I., Maenhaut, W., Marsden, N., Saarikoski, S.
K., Schnelle-Kreis, J., Surratt, J. D., Szidat, S., Szmigielski, R., and Wisthaler, A.: The molecular identification of organic
compounds in the atmosphere: state of the art and challenges, Chemical reviews, 115, https://doi.org/10.1021/cr5003485, 2015.

Pant, P. and Harrison, R. M.: Estimation of the contribution of road traffic emissions to particulate matter concentrations from
field measurements: A review, Atmos. Environ., 77, 78–97, https://doi.org/10.1016/j.atmosenv.2013.04.028, 2013.

Peng, Z. and Jimenez, J. L.: Radical chemistry in oxidation flow reactors for atmospheric chemistry research, Chem. Soc. Rev.,
49, 2570–2616, https://doi.org/10.1039/C9CS00766K, 2020.



Pernigotti, D., Belis, C. A., and Spano, L.: SPECIEUROPE: The European data base for PM source profiles, Atmos. Pollut. Res., 7, 307–314, https://doi.org/10.1016/j.apr.2015.10.007, 2016.

Platt, S. M., Haddad, I. E., Pieber, S. M., Zardini, A. A., Suarez-Bertoa, R., Clairotte, M., Daellenbach, K. R., Huang, R.-J., Slowik, J. G., Hellebust, S., Temime-Roussel, B., Marchand, N., Gouw, J. de, Jimenez, J. L., Hayes, P. L., Robinson, A. L., Baltensperger, U., Astorga, C., and Prévôt, A. S. H.: Gasoline cars produce more carbonaceous particulate matter than modern filter-equipped diesel cars, Sci Rep, 7, 1–9, https://doi.org/10.1038/s41598-017-03714-9, 2017.

Presto, A. A., Gordon, T. D., and Robinson, A. L.: Primary to secondary organic aerosol: evolution of organic emissions from 705 mobile combustion sources, Atmospheric Chemistry and Physics, 14, 5015–5036, https://doi.org/10.5194/acp-14-5015-2014, 2014.

R'Mili, B., Boréave, A., Meme, A., Vernoux, P., Leblanc, M., Noël, L., Raux, S., and D'Anna, B.: Physico-Chemical Characterization of Fine and Ultrafine Particles Emitted during Diesel Particulate Filter Active Regeneration of Euro5 Diesel Vehicles, Environ. Sci. Technol., 52, 3312–3319, https://doi.org/10.1021/acs.est.7b06644, 2018.

Robinson, A. L., Donahue, N. M., Shrivastava, M. K., Weitkamp, E. A., Sage, A. M., Grieshop, A. P., Lane, T. E., Pierce, J. R., and Pandis, S. N.: Rethinking Organic Aerosols: Semivolatile Emissions and Photochemical Aging, Science, 315, 1259–1262, https://doi.org/10.1126/science.1133061, 2007.

Rogge, W. F., Hildemann, L. M., Mazurek, M. A., Cass, G. R., and Simoneit, B. R. T.: Sources of fine organic aerosol. 2. Noncatalyst and catalyst-equipped automobiles and heavy-duty diesel trucks, Environ. Sci. Technol., 27, 636–651, 715 https://doi.org/10.1021/es00041a007, 1993.

Röhler, L., Schlabach, M., Haglund, P., Breivik, K., Kallenborn, R., and Bohlin-Nizzetto, P.: Non-target and suspect characterisation of organic contaminants in Arctic air – Part 2: Application of a new tool for identification and prioritisation of chemicals of emerging Arctic concern in air, Atmospheric Chemistry and Physics, 20, 9031–9049, https://doi.org/10.5194/acp-20-9031-2020, 2020.

Röhler, L., Bohlin-Nizzetto, P., Rostkowski, P., Kallenborn, R., and Schlabach, M.: Non-target and suspect characterisation of organic contaminants in ambient air – Part 1: Combining a novel sample clean-up method with comprehensive two-dimensional gas chromatography, Atmospheric Chemistry and Physics, 21, 1697–1716, https://doi.org/10.5194/acp-21-1697-2021, 2021.

Rostkowski, P., Haglund, P., Aalizadeh, R., Alygizakis, N., Thomaidis, N., Arandes, J. B., Nizzetto, P. B., Booij, P., Budzinski, 725 H., Brunswick, P., Covaci, A., Gallampois, C., Grosse, S., Hindle, R., Ipolyi, I., Jobst, K., Kaserzon, S. L., Leonards, P., Lestremau, F., Letzel, T., Magnér, J., Matsukami, H., Moschet, C., Oswald, P., Plassmann, M., Slobodnik, J., and Yang, C.: The strength in numbers: comprehensive characterization of house dust using complementary mass spectrometric techniques, Anal Bioanal Chem, 411, 1957–1977, https://doi.org/10.1007/s00216-019-01615-6, 2019.



Schneider, E., Czech, H., Hartikainen, A., Hansen, H. J., Gawlitta, N., Ihalainen, M., Yli-Pirilä, P., Somero, M., Kortelainen, M., Louhisalmi, J., Orasche, J., Fang, Z., Rudich, Y., Sippula, O., Rüger, C. P., and Zimmermann, R.: Molecular composition of fresh and aged aerosols from residential wood combustion and gasoline car with modern emission mitigation technology, Environ. Sci.: Processes Impacts, 26, 1295–1309, https://doi.org/10.1039/D4EM00106K, 2024.

Schulte, J. K., Fox, J. R., Oron, A. P., Larson, T. V., Simpson, C. D., Paulsen, M., Beaudet, N., Kaufman, J. D., and Magzamen, S.: Neighborhood-Scale Spatial Models of Diesel Exhaust Concentration Profile Using 1-Nitropyrene and Other Nitroarenes, Environ. Sci. Technol., 49, 13422–13430, https://doi.org/10.1021/acs.est.5b03639, 2015.

Simon, H., Beck, L., Bhave, P. V., Divita, F., Hsu, Y., Luecken, D., Mobley, J. D., Pouliot, G. A., Reff, A., Sarwar, G., and Strum, M.: The development and uses of EPA's SPECIATE database, Atmos. Pollut. Res., 1, 196–206, https://doi.org/10.5094/APR.2010.026, 2010.

Sonntag, D. B., Bailey, C. R., Fulper, C. R., and Baldauf, R. W.: Contribution of Lubricating Oil to Particulate Matter Emissions from Light-Duty Gasoline Vehicles in Kansas City, Environ. Sci. Technol., 46, 4191–4199, https://doi.org/10.1021/es203747f, 2012.

Srimuruganandam, B. and Nagendra, S. M. S.: Source characterization of PM10 and PM2.5 mass using a chemical mass balance model at urban roadside, Sci. Total Environ., 433, 8–19, https://doi.org/10.1016/j.scitotenv.2012.05.082, 2012.

Srivastava, D., Tomaz, S., Favez, O., Lanzafame, G. M., Golly, B., Besombes, J.-L., Alleman, L. Y., Jaffrezo, J.-L., Jacob, V., Perraudin, E., Villenave, E., and Albinet, A.: Speciation of organic fraction does matter for source apportionment. Part 1: A one-year campaign in Grenoble (France), Science of The Total Environment, 624, 1598–1611, https://doi.org/10.1016/j.scitotenv.2017.12.135, 2018.

Srivastava, D., Favez, O., Petit, J.-E., Zhang, Y., Sofowote, U. M., Hopke, P. K., Bonnaire, N., Perraudin, E., Gros, V., Villenave, E., and Albinet, A.: Speciation of organic fractions does matter for aerosol source apportionment. Part 3: Combining off-line and on-line measurements, Sci. Total Environ., 690, 944–955, https://doi.org/10.1016/j.scitotenv.2019.06.378, 2019.

Srivastava, D., Daellenbach, K. R., Zhang, Y., Bonnaire, N., Chazeau, B., Perraudin, E., Gros, V., Lucarelli, F., Villenave, E., Prevot, A. S. H., El Haddad, I., Favez, O., and Albinet, A.: Comparison of five methodologies to apportion organic aerosol sources during a PM pollution event, Sci. Total Environ., 757, 143168, https://doi.org/10.1016/j.scitotenv.2020.143168, 2021.

Suarez-Bertoa, R., Zardini, A. A., Platt, S. M., Hellebust, S., Pieber, S. M., Haddad, I. E., Temime-Roussel, B., Baltensperger, U., Marchand, N., Prevot, A. S. H., and Astorga, C.: Primary emissions and secondary organic aerosol formation from the exhaust of a flex-fuel (ethanol) vehicle, Atmospheric Environment, 117, 200–211, https://doi.org/10.1016/j.atmosenv.2015.07.006, 2015.





Thoma, M., Bachmeier, F., Gottwald, F. L., Simon, M., and Vogel, A. L.: Mass spectrometry-based *Aerosolomics*: a new approach to resolve sources, composition, and partitioning of secondary organic aerosol, Atmospheric Measurement

Techniques, 15, 7137–7154, https://doi.org/10.5194/amt-15-7137-2022, 2022.

Thornhill, D. A., Williams, A. E., Onasch, T. B., Wood, E., Herndon, S. C., Kolb, C. E., Knighton, W. B., Zavala, M., Molina, L. T., and Marr, L. C.: Application of positive matrix factorization to on-road measurements for source apportionment of diesel- and gasoline-powered vehicle emissions in Mexico City, Atmospheric Chemistry and Physics, 10, 3629–3644, https://doi.org/10.5194/acp-10-3629-2010, 2010.

Tkacik, D. S., Lambe, A. T., Jathar, S., Li, X., Presto, A. A., Zhao, Y., Blake, D., Meinardi, S., Jayne, J. T., Croteau, P. L., and Robinson, A. L.: Secondary Organic Aerosol Formation from in-Use Motor Vehicle Emissions Using a Potential Aerosol Mass Reactor, Environ. Sci. Technol., 48, 11235–11242, https://doi.org/10.1021/es502239v, 2014.

Tutuianu, M., Bonnel, P., Ciuffo, B., Haniu, T., Ichikawa, N., Marotta, A., Pavlovic, J., and Steven, H.: Development of the World-wide harmonized Light duty Test Cycle (WLTC) and a possible pathway for its introduction in the European legislation,

Transport. Res. Part D-Transport. Environ., 40, 61–75, https://doi.org/10.1016/j.trd.2015.07.011, 2015.

Viana, M., Kuhlbusch, T. a. J., Querol, X., Alastuey, A., Harrison, R. M., Hopke, P. K., Winiwarter, W., Vallius, A., Szidat, S., Prevot, A. S. H., Hueglin, C., Bloemen, H., Wahlin, P., Vecchi, R., Miranda, A. I., Kasper-Giebl, A., Maenhaut, W., and Hitzenberger, R.: Source apportionment of particulate matter in Europe: A review of methods and results, J. Aerosol. Sci., 39, 827–849, https://doi.org/10.1016/j.jaerosci.2008.05.007, 2008.

Vogel, A. L., Lauer, A., Fang, L., Arturi, K., Bachmeier, F., Daellenbach, K. R., Käser, T., Vlachou, A., Pospisilova, V., Baltensperger, U., Haddad, I. E., Schwikowski, M., and Bjelić, S.: A Comprehensive Nontarget Analysis for the Molecular Reconstruction of Organic Aerosol Composition from Glacier Ice Cores, Environ. Sci. Technol., 53, 12565–12575, https://doi.org/10.1021/acs.est.9b03091, 2019.

Wang, Q., He, X., Huang, X. H. H., Griffith, S. M., Feng, Y., Zhang, T., Zhang, Q., Wu, D., and Yu, J. Z.: Impact of Secondary

Organic Aerosol Tracers on Tracer-Based Source Apportionment of Organic Carbon and PM2.5: A Case Study in the Pearl River Delta, China, ACS Earth Space Chem., 1, 562–571, https://doi.org/10.1021/acsearthspacechem.7b00088, 2017.

Wang, Y. and Hopke, P. K.: A ten-year source apportionment study of ambient fine particulate matter in San Jose, California, Atmos. Pollut. Res., 4, 398–404, https://doi.org/10.5094/APR.2013.045, 2013.

Wang, Y., Hopke, P. K., Xia, X., Rattigan, O. V., Chalupa, D. C., and Utell, M. J.: Source apportionment of airborne particulate

matter    using    inorganic    and    organic    species    as    tracers,    Atmos.    Environ.,    55,    525–532, https://doi.org/10.1016/j.atmosenv.2012.03.073, 2012.

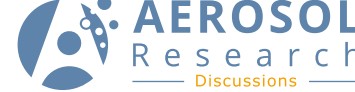

Watson, J., Chow, J., Lowenthal, D., Pritchett, L., Frazier, C., Neuroth, G., and Robbins, R.: Differences in the Carbon Composition of Source Profiles for Diesel-Powered and Gasoline-Powered Vehicles, Atmos. Environ., 28, 2493–2505, https://doi.org/10.1016/1352-2310(94)90400-6, 1994.

Weggler, B. A., Ly-Verdu, S., Jennerwein, M., Sippula, O., Reda, A. A., Orasche, J., Gröger, T., Jokiniemi, J., and Zimmermann, R.: Untargeted Identification of Wood Type-Specific Markers in Particulate Matter from Wood Combustion, Environ. Sci. Technol., 50, 10073–10081, https://doi.org/10.1021/acs.est.6b01571, 2016.

Wong, Y. K., Huang, X. H. H., Louie, P. K. K., Yu, A. L. C., Chan, D. H. L., and Yu, J. Z.: Tracking separate contributions of diesel and gasoline vehicles to roadside $PM_{2.5}$ through online monitoring of volatile organic compounds and $PM_{2.5}$ organic 795 and elemental carbon: a 6-year study in Hong Kong, Atmospheric Chemistry and Physics, 20, 9871–9882, https://doi.org/10.5194/acp-20-9871-2020, 2020.

Worton, D. R., Isaacman, G., Gentner, D. R., Dallmann, T. R., Chan, A. W. H., Ruehl, C., Kirchstetter, T. W., Wilson, K. R., Harley, R. A., and Goldstein, A. H.: Lubricating Oil Dominates Primary Organic Aerosol Emissions from Motor Vehicles, Environ. Sci. Technol., 48, 3698–3706, https://doi.org/10.1021/es405375j, 2014.

Xu, C., Gao, L., Zheng, M., Qiao, L., Wang, K., Huang, D., and Wang, S.: Nontarget Screening of Polycyclic Aromatic Compounds in Atmospheric Particulate Matter Using Ultrahigh Resolution Mass Spectrometry and Comprehensive Two-Dimensional Gas Chromatography, Environ. Sci. Technol., 55, 109–119, https://doi.org/10.1021/acs.est.0c02290, 2021.

Yusuf, A. A. and Inambao, F. L.: Effect of cold start emissions from gasoline-fueled engines of light-duty vehicles at low and high ambient temperatures: Recent trends, Case Studies in Thermal Engineering, 14, 100417, 805 https://doi.org/10.1016/j.csite.2019.100417, 2019.

Zervas, E. and Bikas, G.: Impact of the Driving Cycle on the NOx and Particulate Matter Exhaust Emissions of Diesel Passenger Cars, Energy Fuels, 22, 1707–1713, https://doi.org/10.1021/ef700679m, 2008.

Zhao, Y., Saleh, R., Saliba, G., Presto, A. A., Gordon, T. D., Drozd, G. T., Goldstein, A. H., Donahue, N. M., and Robinson, A. L.: Reducing secondary organic aerosol formation from gasoline vehicle exhaust, PNAS, 114, 6984–6989, 810 https://doi.org/10.1073/pnas.1620911114, 2017.

Zielinska, B., Sagebiel, J., McDonald, J. D., Whitney, K., and Lawson, D. R.: Emission Rates and Comparative Chemical Composition from Selected In-Use Diesel and Gasoline-Fueled Vehicles, Journal of the Air & Waste Management Association, 54, 1138–1150, https://doi.org/10.1080/10473289.2004.10470973, 2004.