# Peer review of "Differentiating between Euro 5 gasoline and diesel light-duty engine primary and secondary particle emissions using multivariate statistical analysis of high-resolution mass spectrometry (HRMS) fingerprint"

_Aerosol Research, 2025_

## Author Comment (AC1)

INE-RIS

maîtriser le risque pour un développement durable

Verneuil en Halatte, 22th October 2025

Dear Shahzad Gani,

We thank you very much for your attention to our manuscript ar-2025-25 now entitled "Differentiation of Euro 5 gasoline vs. diesel light-duty engine primary and secondary particle emissions using multivariate statistical analysis of high-resolution mass spectrometry (HRMS) fingerprint". Please find below our point-by-point response to the referees' comments (in blue) concerning this manuscript. We have addressed each of the reviewer's comments and revised the manuscript accordingly. We think that this new version can now fully meet the Aerosol Research's standards.

With very best wishes,

Dr. Alexandre ALBINET, on behalf of all co-authors INERIS

Parc Technologique ALATA – BP 2, Verneuil-en-Halatte, 60550 (France)

Email: alexandre.albinet@gmail.com, alexandre.albinet@ineris.fr

1

We would like to thank the anonymous referees for their constructive comments on our work. All comments have been considered, and the manuscript is now improved. Changes made are highlighted in orange in the marked versions of the revised manuscript and supplementary material.

**REVIEWER #1**

**Overview**

Noblet et al. studied the primary and secondary emissions of one gasoline vehicle and one diesel vehicle. They applied high-resolution mass spectrometry and three multivariate statistical analysis techniques to the dataset, consisting of filter samples from the vehicles' tailpipe emissions. They found some potential markers for distinguishing emissions from diesel and gasoline vehicles for both primary and secondary organic aerosols. In addition, they have measured several variables continuously during the driving cycles driven.

Overall, the manuscript is well written and provides a detailed description of the research performed. However, as I'm not an expert in filter sampling, in my review I focused on online analysis and the results obtained from the statistical analysis. I feel that analysis and results have been described well and the methods used are well-suited. Overall, the manuscript fits well into the scope of the Aerosol research journal.

I don't have any major concerns related to the manuscript. I have some minor comments and technical corrections regarding the manuscript that I think should be assessed before publication.

**Specific comments**

L299: Was the number of experiments constant for each cycle, i.e. two times for WLTC and four times for MW and Urban? That could be mentioned either in the figure caption or in the Figure itself, to help the reader.

Either for the diesel or the gasoline vehicle, the number of experiments for each cycle was constant: two replicates for the WLTC cycle and four replicates for the hot-start CADC cycles (urban and motorway). To help the reader, the information has been added in the caption of Figure 3 (now) (line 301).

➤ L303: "... notable increase in POA emissions ..." Was the increase indeed in POA emission mass, or is SOA formation included as well? I.e. should it be just "notable increase in emissions"?

The Figure 3 (now) presents the OM average emissions factors for the primary (POA) and secondary (SOA + aged POA) fractions, obtained for the different driving cycles. The purpose of this figure is to show the large increase of the SOA fraction compared to the POA fraction, independently of the vehicle and the driving cycle. In the previous version of the manuscript,

the sentence line 305 mentioned the POA emissions but it was a mistake, as the authors wanted to highlight the significant increase of the SOA fraction. This has been corrected in this version.

Conclusions section: Generalizability of the results. As the results are based on two vehicles, one might think that the variability in vehicles, fuel, and lubricants might affect the distinctiveness of factors in larger dataset. I would appreciate it if the authors could discuss the generalizability of the results in the paper a little bit more than what they have already done in the Conclusions section. Besides the things authors mentioned, my main concerns are related to differences in e.g. lubricant oils and fuels.

The conclusion has been improved to discuss about generatability of the results (line 445): "While these findings are promising and demonstrate the potential of non-targeted HRMS approaches in source apportionment of vehicular OA, the generalizability of the results should be interpreted with caution. The study involved only two vehicles, and variations in vehicle technology, fuel formulation, and lubricant composition are expected to influence the emitted organic aerosol profiles and the distinctiveness of the identified factors. Differences in lubricant additives and fuel aromatic content may alter both the primary emission composition and the pathways of secondary aerosol formation. Future studies should incorporate a larger and more diverse vehicle set, covering a wider range of fuels, lubricants, and emission control technologies, along with increased sampling replicates and oxidation conditions. Such efforts will improve the statistical robustness and environmental relevance".

Figures S10-S12: Especially the bottom-left subplot might be problematic for colorblind people. Could the line styles (e.g. solid, dashed, dotted) also be different for the lines in the same subplot?

These figures have been updated accordingly.

**Technical corrections**

- L168: EFOM ->*EFOM*

**Corrected.**

- L216: Both EIS and IIS are introduced second time, these introductions are unnecessary. Probably you've just forgotten to delete these after the earlier introduction of terms has been added to subsection 2.4.

**Modified**

- L245: What is the meaning of X or Y in R2X? In the text, a letter connected to R2 is X or Y and for Q2 it is constantly Y. In the supplement, the markings are R2X and Q2X.

It was a mistake. To avoid confusion, the letters X or Y have been deleted.

- L301: SOA is also introduced already in the introduction (L44).

**Modified**

- L322: "reproducib ility" (extra space in the text)

**Corrected.**

- Table S9: Rightmost column in QToF section of the Table is dropped down by ½ row, probably because the text of the cell is aligned to the middle. Now it is not completely clear that for which rows the values in the rightmost column are referring.

We did not find any problems in the table. Although, the table has been adjusted to avoid any offset.

**REVIEWER #2**

**Overview**

The article titled "Discrimination of Euro 5 gasoline vs. Diesel light-duty engine primary and secondary particle emissions using multivariate statistical analysis of high-resolution mass spectrometry (HRMS) fingerprint" investigated the chemical composition of primary and secondary organic aerosol emissions from gasoline and diesel motor vehicles and attempted to identify unique chemical fingerprints that could help isolate the ambient contributions of these sources. The study focuses on a highly relevant and contemporary topic. Differentiating between the ambient contributions of gasoline vs diesel vehicles is challenging, yet need of the hour for many regions of the world where these sources dominate air pollution problems. The work employed a broad range of analytical instrumentation to capture the emissions composition in detail, which supports the objectives of this study.

Still, in my view the presented analyses somehow does not make sufficient use of the rich molecular information obtained from this diverse set of instrumentation. I consider this aspect a major shortcoming of this work in its present form. Most of the main text graphs are largely statistical in nature, mandate referring the SI to understand fully and are more suitable for the SI in general. It is difficult for a reader to make use of the main text graphs as standalone visuals to inform their own work, capture the rest of the study or assess it against other work. Yet, given the significance of the research topic and detailed measurements, I think the study can be considered for publication after this and other major concerns detailed below are resolved.

This work represents a proof of concept and an evaluation of the potential of the applied approach. The primary objective was to determine whether potential organic compounds could be identified to distinguish between gasoline and diesel vehicles, based on both their primary and secondary (aged) emissions. The overall organic chemical fingerprints of each sample were compared to highlight potential markers specific to each vehicle type. Some of these markers may correspond to compounds with relatively low signal intensities in the mass spectrometer compared to other more abundant molecules, which explains why they were not fragmented during the analytical run. To enable their identification, MS/MS spectra would be required, typically obtained through targeted re-analysis of the samples. However, compound identification was not part of the initial objectives of this study, which explains, for instance, the absence of MS/MS data for LC-QToF analyses. The identification of these markers remains particularly challenging since many of them are not referenced in LC-associated MS databases. A dedicated investigation would be necessary to elucidate their structure, which was beyond the scope of the present work. The overall mass spectra provide limited informative value; therefore, we have only reported in this manuscript the molecular information with the highest confidence level and scientific relevance, to avoid overloading the text. We acknowledge that readers may find it difficult to navigate through all the datasets and figures presented. Nonetheless, it was not feasible to include all graphs in the main text. We thus selected and presented only the most representative and informative figures, specifically those best illustrating the differentiation between primary and secondary emissions from the two vehicle types.

➤ I find it quite curious that no mass spectra are presented in this study even though an ACSM and QTOF are employed. It would be good to see how the POA and SOA mass spectra looked for different test conditions pre and post PAM oxidation. With comprehensive measurements performed, I think there is value in also showing how similar were these emissions at the very molecular level. This can inform future source apportionment studies working with soft ionization-based measurements (e.g. EESITOF).

ACSM data were primarily used to determine emission factors for both primary and secondary emissions from each vehicle. These results are summarized in Figure 3 (main text) and Figures S12 and S13 (SM). The ACSM data were not used to compare POA and SOA mass spectra, since comprehensive chemical fingerprinting was already achieved using LC- and GC-QToF analyses. Moreover, ACSM mass spectra are known to be highly fragmented, making them unsuitable for detailed molecular comparison.

Regarding the QToF data, only GC-QToF mass spectra were presented. This choice is justified by the type of ionization source used in GC-QToF (electron ionization, EI), which induces extensive fragmentation and thus provides valuable structural information for unknown compounds. An additional advantage of GC-QToF analysis lies in the availability of extensive mass spectral libraries (e.g., NIST), which facilitate compound identification. In contrast, LC-QToF employs an electrospray ionization (ESI) source, which is a soft ionization technique that produces limited fragmentation (aside from minor in-source processes). Consequently, in the initial stage of non-target screening, MS spectra obtained under these conditions are generally not informative, as only a few features are fragmented. A dedicated re-analysis would be required to specifically target and fragment each highlighted marker. However, as compound identification was not the primary objective of this study, MS/MS spectra for the detected markers are scarce.

Finally, providing mass spectra from soft ionization—based techniques such as EESI-TOF would not be relevant here, as the analytical process differ. In EESI-TOF, the goal is to generate Na+ adducts, whereas in LC-ESI-QToF-MS, in addition to get a chromatographic separation that does exist for on-line MS, the focus is on obtaining [M+H]+ or [M–H]- ions while minimizing the formation of Na+ adducts.

The marker species shown as chromatogram peaks in figures S23-31 are not very useful, though the snapshots of the mass spectra are. The chromatogram peak shapes and retention times can vary based on instrument settings and column type used for separation. These figures should be revised to make the mass spectral peaks more prominent.

The specific ion chromatograms corresponding to the various markers identified in the analysed samples are presented in Figures S22–S30. The presence of a distinct peak in the chromatogram indicates the occurrence of the corresponding marker in each sample, whereas the absence of a peak confirms that the marker is not present. This approach allowed us to validate the presence of specific markers for each vehicle type, independently of the driving conditions. Notably, certain potential markers were detected only under specific driving conditions. These were therefore excluded from further consideration.

In figures S10-S12, it is very difficult to assess how different variables correlate. Scatter plots alongside different timeseries would be helpful.

These figures have been updated but scatter plots have not been included because it is not relevant for the purpose of the work presented here and for the discussion.

> Writing can be improved at several instances in the manuscript and better proofread.

The manuscript has been carefully proofread, and several improvements have been made.

The research objectives should be made sharper. For example, in line 250, the authors state that marker identification was not the primary aim of this work. However, line 85 contradicts this statement in goal-setting where revealing the differentiating markers for POA and SOA is noted as the objective of this work. The word "fingerprint" is literally in the title of this manuscript.

The aim of this work was to combine a non-target screening (NTS) approach, designed to detect as many molecular features as possible, with multivariate statistical analyses to explore, for instance, potential source markers of particulate matter (PM). It is important to distinguish between highlighting potential markers and identifying them, as the latter requires dedicated and targeted analytical work.

To clarify this point, we have revised the main text (lines 84–85) as follows: "The objective of this work was to assess the potential of NTS analyses using HRMS data to highlight differential chemical features (markers) of primary organic aerosols (POA) and SOA from gasoline and diesel vehicles".

Similarly, I had to sift through the manuscript text continuously to link the discussion of results with the order in which the experiments were conducted. I think having a few timeseries (e.g. ACSM organics and a few others) systematically presenting the different phases (test conditions) of the experiments in the main text would have helped.

A time series showing primary particle number, NOx and CO concentrations at emission for the studied EURO 5 diesel and gasoline vehicles during the ambient start WLTC driving cycle, has been added to the main text (Figure 2).

Line 387: What does "less marked" mean? Does it mean there are less differentiating markers?

The primary objective of the PCA was to provide a macroscopic and synthetic overview of the correlations between variables, without imposing relationships that may not exist. PCA allowed us to visualize these correlations and to identify the main distinct groups corresponding to diesel and gasoline SOA. The key point we intended to convey with this sentence was that the differentiation between the two sample groups (diesel and gasoline SOA) was less pronounced than that observed for the POA samples. This observation suggests that the chemical compositions associated with the variables for each group are more similar in the case of SOA. It does not, however, imply that fewer features were detected in the analysis of secondary fractions compared with primary ones.

We agree with the reviewer's comment that the initial phrasing may have been confusing. Accordingly, the sentence was revised as follows (line 388): "This separation was not as clear for the ESI(+) LC-QToF data, suggesting that the diesel and gasoline SOA chemical fingerprints could be more closely related than those observed for POA compounds".

Another example of a confusing sentence is lines 325 - 327: I am not sure what the authors mean by "(trans-)formation state of the OA" that distinguishes between primary and secondary emissions.

Secondary organic aerosols can be obtained by the formation of new particles or by the reaction of existing organic species from primary organic aerosols (oxidized POA). So, the term used here is appropriate.

The use of acronyms/identifiers e.g. LC NEG SOA G-1 is very inconvenient, and forces the reader to oscillate between the main text and the SI.

We acknowledge that these acronyms may be confusing. However, we considered it important for readers to be informed of the analytical origin of each marker (LC(ESI+), LC(ESI-) or GC) and to know whether it was specific to the primary or secondary fraction. To avoid oscillating between the main text and the SI, Table S12 has been moved into the main text (Table 1 now).

Line 390: "The high NOx levels may have also triggered distinct atmospheric reactions, leading to the formation of nitrogen-containing compounds." I reckon the authors are referring to the formation of organo-nitrates but they should consider the branching ratios before suggesting this to be important and that the ratios vary by the tertiary, secondary or primary nature of carbon in precursor molecules. To a large extent, high NOx conditions direct reactions toward specific pathways by converting peroxy- to alkoxy- radicals resulting in dominant production of oxygenated species. If a speculation is mandatory here, I recommend reinforcing it more effectively.

This suggestion has been integrated in the text as follows (line 391): "This observation aligns with the elevated  $NO_x$  concentrations measured during this driving cycle. Under high- $NO_x$

conditions, the reaction of peroxy-radicals with  $NO_x$  can yield either alkoxy radicals or organic nitrates, depending on the branching ratio, which varies with the degree of carbon substitution in the precursor VOCs. While the formation of organonitrates may contribute to nitrogen-containing SOA, high  $NO_x$  levels generally favour fragmentation pathways leading to more oxygenated, volatile species (Gentner et al., 2017; Pullinen et al., 2020). Overall, the high  $NO_x$  levels may have triggered distinct atmospheric reactions, leading to the formation of nitrogen-containing compounds or more oxygenated species.

➤ The panels are not labeled in figures 2 and 5. The axes are not labeled for some panels in figure 5.

The panels of the Figure 3 (now) are well labelled. Some other information has been added in the caption as follows: "Figure 3. Diesel (top) and gasoline (bottom) Euro 5 OM average emissions factors for the primary (POA) and secondary (SOA + aged POA) fractions, average NOx concentrations after dilution exhaust and average OH exposures obtained with the ambient start WLTC, hot-start CADC motorway (MW) and urban driving cycles. The error bars correspond to the standard deviation ( $\pm$  2  $\sigma$ ) for the number of experiments performed (n = 2 for the WLTC cycle and n= 4 for the hot-start CADC MW and urban cycles)."

Concerning the Figure 6 (now), the caption enables to explain the different panels. However, the axis labels were effectively missing. This has been corrected.

Figure 4: The hierarchical structure is not discernable. I wonder how useful is this figure in the main text.

The purpose of HCAs is to show, in a macroscopic way, the links between different samples based on their features. In this case, it is interesting to highlight that: 1) primary aerosol samples are chemically closer to each other than secondary aerosol samples, which are themselves chemically closer to each other and 2) there are groups of features that are present in some groups and completely absent in others.

Lines 424 - 426: I am not sure how the authors declared some of these points (e.g. red circle on the left) to be an outlier/erroneous. More supporting evidence should be provided in this regard. An erroneous status cannot be accorded to a datapoint simply by the virtue of its positioning on the VK plot. There are multiple points on the graph where POA and SOA nearly overlap and these are treated as valid data.

We have modified the text to only use the terms questionable or doubtful to avoid any misunderstanding in the interpretation of the results (lie 428): "However, a few questionable molecular formulas were identified (highlighted in Figure 7 and listed in Table 1) which fall outside the typical elemental composition patterns and are likely doubtful and should be considered with caution."

**Minor comments:**

- The word "Discrimination" in the title is not accurate. Discrimination represents preference or bias toward one over the other. I think what the authors are looking for is along the lines of differentiation or distinction. I suggest revising the title.

We have revised the title as follows: "Differentiation of Euro 5 gasoline vs. Diesel light-duty engine primary and secondary particle emissions using multivariate statistical analysis of high-resolution mass spectrometry (HRMS) fingerprint".

- Table S3 does not include offline instrumentation even though the table caption says "all".

**This has been corrected.**

- The word "Diesel" should have a small "d", i.e. diesel, everywhere in the text.

**This has been corrected.**

Line 43: consider replacing "photochemical" with "oxidation".

**Modified**

Line 142: closing parenthesis is missing.

**This has been corrected.**

- Line 302: The ranges overlap. Thus the sentence needs to be revised. Stating the ratio between averages of the two ranges makes more sense with ranges (xy - yx vs. xx - yy) in brackets.

The sentence has been modified (line 303).

- Line 322: remove the space from the word "reproducibility".

**This has been corrected.**

Figure 3: should the legend have filled circles with a black outline?

It is understandable as it is currently.